# Soil Application of Nano Silica on Maize Yield and Its Insecticidal Activity Against Some Stored Insects After the Post-Harvest

**DOI:** 10.3390/nano10040739

**Published:** 2020-04-12

**Authors:** Mehrez E. El-Naggar, Nader R. Abdelsalam, Moustafa M.G. Fouda, Marwa I. Mackled, Malik A.M. Al-Jaddadi, Hayssam M. Ali, Manzer H. Siddiqui, Essam E. Kandil

**Affiliations:** 1Pre-Treatment and Finishing of Cellulosic based Fibers Department, Textile Industries Research Division, National Research Center, 33 El- Behooth St, Dokki, Giza 12311, Egypt; 2Agricultural Botany Department, Faculty of Agriculture, Saba Basha, Alexandria University, Alexandria P.O. Box 21531, Egypt; 3Department of Stored Product Pests, Plant Protection Institute, Agriculture Research Center (ARC), Sabahia, Alexandria P.O. Box 21616, Egypt; Mazennour2@yahoo.com; 4Ministry of Commerce, Trade and Financial Control Department, Trade Control Department, Division Quality Control, Baghdad 13201, Iraq; malekmutalk@gmail.com; 5Botany and Microbiology Department, College of Science, King Saud University, P.O. Box 2455, Riyadh 11451, Saudi Arabia; hayhassan@ksu.edu.sa (H.M.A.); mhsiddiqui@ksu.edu.sa (M.H.S.); 6Timber Trees Research Department, Sabahia Horticulture Research Station, Horticulture Research Institute, Agriculture Research Center, Alexandria 21526, Egypt; 7Plant Production Department, Faculty of Agriculture (Saba Basha), Alexandria University, Alexandria P.O. Box 21531, Egypt; essam.kandil@alexu.edu.eg

**Keywords:** maize, NPK, SiO_2_-NPs, productivity, fertilizer, mineral, weevils, LC_50_, toxicity

## Abstract

Maize is considered one of the most imperative cereal crops worldwide. In this work, high throughput silica nanoparticles (SiO_2_-NPs) were prepared via the sol–gel technique. SiO_2_-NPs were attained in a powder form followed by full analysis using the advanced tools (UV-vis, HR-TEM, SEM, XRD and zeta potential). To this end, SiO_2_-NPs were applied as both nanofertilizer and pesticide against four common pests that infect the stored maize and cause severe damage to crops. As for nanofertilizers, the response of maize hybrid to mineral NPK, “Nitrogen (N), Phosphorus (P), and Potassium (K)” (0% = untreated, 50% of recommended dose and 100%), with different combinations of SiO_2_-NPs; (0, 2.5, 5, 10 g/kg soil) was evaluated. Afterward, post-harvest, grains were stored and fumigated with different concentrations of SiO_2_-NPs (0.0031, 0.0063. 0.25, 0.5, 1.0, 2.0, 2.5, 5, 10 g/kg) in order to identify LC_50_ and mortality % of four common insects, namely *Sitophilus oryzae, Rhizopertha dominica, Tribolium castaneum,* and *Orizaephilus surinamenisis*. The results revealed that, using the recommended dose of 100%, mineral NPK showed the greatest mean values of plant height, chlorophyll content, yield, its components, and protein (%). By feeding the soil with SiO_2_-NPs up to 10 g/kg, the best growth and yield enhancement of maize crop is noticed. Mineral NPK interacted with SiO_2_-NPs, whereas the application of mineral NPK at the rate of 50% with 10 g/kg SiO_2_-NPs, increased the highest mean values of agronomic characters. Therefore, SiO_2_-NPs can be applied as a growth promoter, and in the meantime, as strong unconventional pesticides for crops during storage, with a very small and safe dose.

## 1. Introduction

The global population will rise to 9 billion by the year of 2050, and the existing agricultural practices cannot satisfy this growing demand for food without variations in the fertilizer’s application. Nanotechnology is currently being applied in abundant fields such as medicine, pharmaceutics, electronics, and agriculture. The size and purity of nanomaterials results significantly in various procedures as well as improvements in the physical and chemical properties of any materials due to their small size which in turn, caused very large surface area [1,2].

Worldwide, *Zea mays* L. is considered as one of the most important cereal crops [3,4,5]. The area of maize cultivation in Egypt is 1.1 million hectares (average yield about 7.4 t/ha) and in the world 188 million hectares (an average yield about 5.6 t/ha) reported by (FAO, 2007). 

Elements such as Nitrogen, Phosphor, and Potassium, abbreviated as NPK, are considered vital macronutrients for meristematic production and several physiological processes in plant [6,7,8,9,10,11,12,13,14,15] for instance, shoot, root system, flowers etc., moreover, leading to effective water translocation and nutrition, improve the process of photosynthesis [16]. On the other hand, silicon can be considered as a micro nutrient and it is supportive for plant growth, mainly in dry environments, in order to hold water and bind other nutrients, in addition to increasing the cell strength [17]. Moreover, the utilization of silicon makes the plant shoot system more erect as the effect of a high dose of nitrogen fertilizers, which will improve plant photosynthesis, chlorophyll content, and product quality [18,19] evaluated the effectiveness of nano fertilizers relative to their conventional analogues and the results displayed that nano fertilizers has the largest increase in median efficacy increase (29%). Thus, using SiO_2_-NPs, as nanofertilizer together with NPK will increase the absorbability of fertilizers by plants and, hence, it will be more effective than conventional chemical fertilizers [20]. Prihastanti et al. [21] noticed that SiO_2_-NPs are an important nanofertilizer which contains silicon which is essential to the monocotyl plants, such as maize, to increase the growth and productivity as well, rather than, NPK alone, that comprises N, P, and K (macronutrient). The combination between NPK and SiO_2_-NPs limits the utilization of hazard chemical fertilizers besides its capability to improve maize production [22,23]. 

There are many ways that extensively used for the production of silica nanoparticles (SiO_2_-NPs) such as electrochemical, hydrothermal, plasma–metal hydrogen reaction, micro-emulsion, arc discharge, chemical vapor condensation, vapor phase laser pyrolysis, radiation, sonication, laser, biological, and chemical methods [12,24,25,26,27,28,29,30,31,32,33,34,35,36]. One of these chemical methods is the sol–gel process which is extensively used in order to produce homogenous silica products in a powder form. The produced silica gels are non-toxic and suitable to be used for several domains particularly, agricultural applications.

As far as post-harvest is concerned, maize grains are considered one of the identical hosts for many stored products insects such as, *Sitophilus* and *Rhizopertha dominica*, *Tribolium castaneum,* and *Orizaephilus surinamenisis*, which resulted in a loss of more than 25%. Fumigants and residual pesticides are widely used to protect the stored grains from infestation by plague [37].

Hereby, this current research work aimed to prepare silica nanoparticles (SiO_2_-NPs) in high concentration with small size and distribution to be used as an alternative and effective nanomaterial for the protection of stored grains. It is expected that these nanoparticles will reduce the utilization of hazardous chemical pesticides which, in turn, will reduce the health hazard from residual toxicity. Additionally, the prepared SiO_2_-NPs enhances to solve the insect resistance to the conventional insecticides (phosphine and pyrethroids) too [38,39,40,41].

Overall, the main objectives of this current research are dived into three key subjects: a) preparation and characterization of SiO_2_-NPs using the sol–gel method, b) evaluation of the influence of the combination between SiO_2_-NPs and mineral NPK, as soil application and their interaction with the plant characteristics of maize, and c) application of SiO_2_-NPs as an alternative pesticide to combat pests infested maize grains through post-harvest, as well as to resolve insect resistance to the conventional pesticides.

## 2. Materials and Methods 

### 2.1. Experiment Place and Design

The present investigation was carried out at the Experimental Farm, Faculty of Agriculture (Saba Basha), Alexandria University, Alexandria, Egypt and Department of Stored Product Pests, Agriculture Research Center, Sabahia, Alexandria, Egypt, cooperated with Botany and Microbiology Department, College of Science, King Saud University, Riyadh, Saudi Arabia through the two successive summer seasons of 2018 and 2019, in split plot design with 3 replications. The major plot was mineral NPK fertilizers rates ((0% (0:0:0:0), 50% (144:30:30), and 100% (288:60:60)/ha), while sub plots were allocated by silica NPs concentration (0.0, 2.5, 5.0 and 10.0 g/kg autoclaved soil) in both seasons.

### 2.2. Sol-Gel Synthesis of Silica Nanoparticles (SiO_2_-NPs)

For the preparation of SiO_2_-NPs via sol-gel method, 35 mL of H_2_O was mixed with 65 mL of absolute alcohol for 5 min under mechanical stirring. After that, 25 mL of tetraethyl orthosilicate (TEOS) was added dropwise to the previous ethanol/water solution and kept under mechanical stirring for 60 min at room temperature. To this end, ammonia solution was added dropwise until the complete formation of gel. Thus, it was noted that the solution was converted to gel (sol-gel process). The formed gel was submitted to ultra-centrifugation for 2 h at 7000 rpm. Finally, the precipitated wet gel was collected and washed three times with distilled water in order to remove the undesired/unreacted compound (TEOS). The wet gel was subjected again to ultracentrifugation. At the end, the obtained gel was left for calcination at 700 °C for 5‒7 h.

### 2.3. Physical Characterization of Silica Nanoparticles (SiO_2_-NPs)

The sample for transmission electron microscope (TEM) examination was prepared by placing the dispersed SiO_2_-NPS on a carbon-coated copper grid and left for drying at room temperature before being characterized via TEM instrument (JEOL 200 kV, Tokyo, Japan). The particle size and zeta potential of SiO_2_-NPs in its colloidal solution and after submission for 15 min of sonication were assessed using particle size analyzer (Nano-Sizer SZ90, Malvern instruments Ltd., Cambridge, UK). The size distribution and zeta potential of the as prepared SiO_2_-NPs was measured at pH = 7 and 25 °C. Scanning Electron Microscopy (SEM; JEOL, JSM-6360LA, Tokyo, Japan) instrument was used to investigate the internal structure and surface morphology of SiO_2_-NPS. X-ray diffraction (XRD) analysis was performed to examine the crystallinity and the specific peaks for the formed SiO_2_-NPs using an XRD device (Panalytical Emperian, Istanbul, Turkey) having Cu_Ka_ radiation and operating with 40 kV and a 2-theta range of 10–80.

### 2.4. Soil Characterization and Preparing Materials 

A surface sample of soil (0‒30 cm) was collected before planting to identify some soil physical and chemical properties, as shown in Table 1. According to Keeney et al. [42], the previous crop was clover (berseem) in both seasons. Nano silica powder were mixed well with autoclaved soil and applied at two times, the first time before the first irrigation (after thinning) and the second time before the second irrigation. The recommended dose of NPK as following; the recommended dose of phosphorus fertilizer was used at rate of 60 kg P2O5/ha (where ha = 0.42 feddan) from calcium super phosphate (12.5% P2O5) and potassium rate of 60 kg K2O/ha from potassium sulphate (48% K2O) with soil preparation. The recommended dose of mineral N at the different rate of 288 kg N/ha was fully given in the form of urea (46.5% N) such as previous adding. Each plot size was 12.60 m^2^, included 6 ridges, each 3 m in length and 0.70 m in width, with the distance between hills of 25 cm. The grains of maize hybrid (TWC 1100) were taken from Maize Research Division, Agriculture Research Center, Ministry of Agriculture, Cairo, Egypt. Theses grains were sown on May 15^th^ and 14^th^ of 2018 and 2019 seasons, respectively.

### 2.5. Maize Yield and Yield Compound Characteristics 

The maize yield and yield compound parameters were calculated after harvest and the data were obtained as an average of two ridges from middle of each plot. The protein% was concluded according to the methods of Helrich (1990) by assessing the total nitrogen in the grains and multiplied by 6.25 to obtain the percentage according of grains protein% [43]. 

### 2.6. Post-Harvest Experiment

#### 2.6.1. Insect Culture

Two insects, *Sitophilus oryzae* and *Rhizopertha dominica*, were reared under laboratory conditions (27 ± 1 °C and 65 ± 5% R.H.) using autoclaved maize grains which obtained from the first experiment after storage two months in dry conditions, *T. castaneum* and *O. surinamenisis* was reared on maize flour mixed with yeast (10:1, w/w), in 1-L glass jars, which were covered by fine mesh cloth for ventilation as reported by [44]. Adult insects used in toxicity tests were about 1‒2 weeks old. All investigational procedures were conducted under the identical conditions as culture.

#### 2.6.2. Contact Film Toxicity Bioassay

The previous SiO_2_-NPs was evaluated (2.5, 5, and 10 g/kg) and the mortality percentage was 100%, so we decreased the concentrations to the lower doses for obtaining the LC_50_ to the four stored insects. Toxicity of the nine evaluated SiO_2_-NPs (0.0031, 0.0063, 0.25, 0.5, 1.0, 2.0, 2.5, 5, and 10 g/kg) against the weevils of *S. oryzea, R. domonica, T. castanium,* and *O. surinamenisis* (adults) were examined by transferring 20 adults into glass jars (250 mL) containing 100 g of sterilized maize grains and admixed them well with different doses of SiO_2_-NPs according to the method of Su and Zabik (1972) [45]. Control jars continues maize grains alone. Three replicates were used for all treatment and control. The mortality percentage (M%) were measured according to Finney (1971) after one, two, and three days and LC_50_ values were determined according to the method of [46].

### 2.7. Statistical Analysis

Obtained data were subjected to the proper system of statistical analysis of variance as defined by [47]. The means were compared using L.S.D. test at 5% probability using a split model as found in CoStat 6.311 program, PMB320, Monterey, CA93940, and USA [48].

## 3. Results

In this current research work, it is aimed to develop a new strategy for the soil applications in term of feeding or fertilizing, and at the same time, as pesticide for combating the different kinds of pests that are found through storing maize grains. Nanotechnology in this research work is implemented through the production of silica nanoparticles, SiO_2_-NPs, which serve as enhancement agents for the soil application as well as a pesticide agent in post-harvest, for maize grains during storage for long time. As reported previously in the literature; the production of SiO_2_-NPs is depending on two major chemical steps: the first one is nucleation that occurs by the hydrolysis of tetraethyl orthosilicate to form silanol groups, which is followed by the second step, growth stage, that takes place by the condensation between the silanol groups formed leading to the construction of siloxane bridges (Si–O–Si) that, yield at the end the entire silica nanoparticle formula. The hydrolysis step is carried out in the presence of alkali like ammonia (NH_3_) that acts as a reaction enhancement for the formation of the end product. Scheme 1, represents the preparation of SiO_2_-NPs and their application as soil nanofertilizer for maize, as well as an insecticide for the stored maize insects.

Below is the full analysis for the formed SiO_2_-NPs by means of TEM, particle size analyzer, zeta potential, SEM, and XRD techniques. 

Firstly, TEM was represented for the formed SiO_2_-NPs in order to clarify the particles shape and their distributions. The TEM images of SiO_2_-NPs are taken at three different magnifications to clarify the actual shape of the synthesized particles. Figure 1A‒C shows that the particles shape is spherical with low disparity which may be attributed to the cluster effect of silica particles. However, these aggregated particles are less than 50 nm. 

To confirm the particle size and stability, hydrodynamic average size was examined using dynamic light scattering (DLS) as represented in Figure 1D. It is observed that the average size is around 68 nm. As can be clearly seen, the particle size obtained from DLS is little bit larger than that obtained from TEM figures. This can be claimed in terms of a swelling effect. For the DLS technique, the sample during examination is kept in distilled water for a long time (duration of measurements; 18 run). In this case, the particles are marginally swelled, which, in turn, leads to a slight increase in the size of the examined particles. 

In light of stability of surface charge of the produced SiO_2_-NPs, zeta potential (Figure 1E) was carried out to provide us an information about the particle stability against aggregation. It is well known that value of Zeta potential above +30 mV or ‒30 mV is considered as good stabilized sample and already protected from further aggregation or agglomeration. Thus, the nominated examination is very important to stand out for the sample stability after its preparation. Therefore, the average zeta potential of SiO_2_-NPs is evaluated ad plotted in Figure 2B. It is depicted that the zeta potential value of SiO_2_-NPs is recorded as ‒40 mV. Such a value means that the particles are kept away from further aggregation, even after a long time.

In order to clarify the morphological surface structure of SiO_2_-NPs, the sample was scanned at different magnifications using SEM. The scanned SiO_2_-NPs sample is displayed in Figure 2A,B. As shown in the SEM images, the prepared powder consists of spherical particles with well-defined borders. The calcination process at high temperature (600–700 °C) is an important factor for purification and the formation of particles with spherical morphology and regular shape.

In order to outline the crystallinity and purity of the aforementioned powder sample, X-ray diffraction pattern (XRD) was utilized. XRD analysis was carried about between 2 theta degree (10–80). It is disclosed from Figure 3 that SiO_2_-NPs exhibit three major peaks at 21.88°, 38.5°and 45.9° which correspond to (100), (110), and (201) planes. The obtained peak value is in accordance with that of JCPDS Card #850335 for SiO_2_-NPs. Based on the aforementioned peaks, SiO_2_-NPs can be prepared successfully using the sol–gel technique.

### 3.1. Growth and Yield Compounds

The growth and yield characters, such as leaf chlorophyll content, plant height, ear length, grains number/row, grains number/ear, weight of 100 grains, grain yield, straw yield, biological yield, harvest index, and protein content of maize hybrid were significantly affected by a combination of NPK fertilizers and SiO_2_-NPs concentrations in an average of both 2018 and 2019 seasons as found in Table 2. The results verified that the application of NPK at the recommended dose (RD = 100%) recorded the maximum mean values of leaf chlorophyll content (38.72 SPAD), plant height (195.79 cm), ear length (20.17 cm), grains number/row (41.67 grains/row), grains number /ear (583.33 grains/ear), weight of 100 grains (43.00 g), grains yield (4.79 t/fed), straw yield (6.29 t/fed), biological yield (11.08 t/fed), harvest index (43.23%) and content of protein in grain (10.18%) followed by fertilization by 50% of recommended dose from mineral NPK, while the lowest ones were given by untreated treatment (NPK = 0).

Regarding, effect soil application of SiO_2_-NPs on maize yield and components characters, the results detected that with the increase of SiO_2_-NPs concentrations from 0 up to 10 g/kg, there is an increase in all the studied characters (Table 2). The highest concentration of SiO_2_-NPs verified the greatest mean values of leaf chlorophyll content (40.27 SPAD), plant height (201 cm), ear length (20.78 cm), grains number /row (41.77 grains/row), grains number /ear (583.35 grains/ear), weight of 100- grains (44.56 g), grain yield (5.00 t/fed), straw yield (6.24 t/fed), biological yield (11.24 t/fed), harvest index (44.48%) and protein content in grain (10.37%) followed by the concentration 5 g/kg Nano silica as compared with the other concentration.

Combinations between NPK and SiO_2_-NPs showed a highly significant interaction, as found in Table 2 for all the studied characters. The significant interaction shows that the response of effect of treatments of the first factor dependable on the levels of the other factor.

The results in Table 3 presented the interaction effect between mineral NPK and SiO_2_-NPs, where the highest mean values of chlorophyll content (45.13 SPAD), plant height (222.33 cm), ear length (22.33 cm), grains number/row (44.00 grains/row), grains number /ear (616.00 grains/ear), weight of 100 grains (47.67 g), grain yield (5.59 t/fed), straw yield (7.09 t/fed), biological yield (12.68 t/fed), and content of protein in grain (12.69%) were attained from fertilizing maize plants by the rate of 50% of recommended dose from mineral NPK and soil application of SiO_2_-NPs (10 g/kg) except the highest mean of harvest index (46.30%) recorded with 50% of recommended dose from mineral NPK and 5g/kg SiO_2_-NPs.

In comparison with the other treatments, meanwhile the lowest ones were given with untreated plots (0 NPK + 0 SNPs), that cleared the role of SiO_2_-NPs in the response of maize crop to NPK. The data found in Table 3 demonstrate the interaction impact of mineral NPK fertilizer and SiO_2_-NPs application of some maize characters, where the highest values of the studied characters recorded with 50% recommended dose of mineral NPK + 10 g/kg SiO_2_-NPs.

SiO_2_-NPs with high surface area produced in the commercial way are employed for the growth and productivity of maize as an unconventional source of fertilizer. Physiological transformations that are due to SiO_2_-NPs fertilization considerably increase the growth and yield characters in maize plants.

### 3.2. Toxicity of SiO_2_-NPs against Stored Products Insects 

Nine concentrations of SiO_2_-NPs were appraised against four stored products insects *S. oryzae*, *R. dominica*, *T. castaneum* and *O. surinamenisis* (Figure 4), the initial results obtained by the application of SiO_2_-NPs; 2‒10 g/kg displayed 100% mortality %, thus we decreased the concentrations used to get the LC_50_ for the SiO_2_-NPs and three exposer time. 

Figure 5 shows the difference between healthy and infected maize grains by different stored insects.

The data in Figure 6, Figure 7 and Figure 8 show that, after 24 h of the treatment, *R*. *Dominica* became more sensitive to SiO_2_-NPs followed by *O. surinamenisis;* LC_50_ were 0.336 (range, 0.177‒0.521) and 0.768 (range, 0.438‒1.495) g/kg respectively. While the other LC_50_ was 1.240 (range, 0.995‒1.662) and 1.450 (range, 0.971‒3.290) g/kg registered for *S. Oryzae*, *T. Castañea*. With respect to mortality percentage after 24 h, the lowest concentrations of SiO_2_-NPs such as 0.0031 and 0.0063 were not successful in the case of four species, while the M% increased with concentration changes. 

Compared with LC_50_ of *R. dominica* and *O. surinamenisis* with M%, findings showed that for both insects, these values ranged from 0.25 to 0.5 g/kg and ranged from 11.66–63.3% and 1036.65% respectively (Figure 6, Figure 7 and Figure 8). After 24 h, 2 g/kg SiO_2_- NPs displayed that 100% of mortality for all the species. Meanwhile, 1 g/kg of SiO_2_- NPs showed 86.6% mortality for *R. dominica* comparing with 5% for *S. oryzae*; 10% for *T. castaneum* and 36.65% for *O. surinamenisis*. 

Results for both *R. dominica* and *O. surinamenisis* recorded the lowest LC_50_; 0.014 (range, 0.005–0.035) g/kg and 0.008 (range, 0.004–0.006) g/kg comparing with the other two insects; 0.270 (range, 0.114–0.676) g/kg for *T. castaneum* and 0.388 (range, 0.158–1.087) g/kg for *S. oryzae* (Figure 6, Figure 7 and Figure 8). *S. oryzae* showed 30% mortality % at 1.0 g/kg of SiO_2_-NPs, *R. dominica* (90%); T. castaneum (68.3%) and 100% for *O. surinamenisis*. After 48 h of treatments, the data showed that *O. surinamenisis* was very highly sensitive to the SiO_2_-NPs compared with other insects. 

After 72 h, the effect of SiO_2_-NPs in Figure 6, Figure 7 and Figure 8 impact posed that, *R. dominica* and *O. surinamenisis* recorded the lowest LC_50_ values were 0.002 (range, 0.0004–0.006) and 0.002 (range, 0.0005–0.007) g/kg, while *T. castaneum* (0.034) (range, 0.015–0.072 g/kg) and *S. oryzae* (0.263) (range, 0.055–0.014). From 0.25 to 2.0 g/kg of SiO_2_-NPs, the *O. surinamenisis* displayed 100% mortality %, while *S. oryzae* was more resistance to SiO_2_-NPs which exhibited 93.3% under 1.0 g/kg (Figure 8) compared to the other insects that disclosed 100%.

## 4. Discussion

The main significance of this current work is to prepare silica nanoparticles in a very high concentration using the sol-gel technology. The prepared silica nanoparticles in their current form are not toxic, since our aim in this current work was to prepare them in a pure form without any impurities. Thus, the calcination process has been carried out to degrade the undesired and unreactive compounds of TEOS or ammonia, ethanol substances. The next step is to use this industrial scale up, of silica nanoparticle in agricultural domain, as both growth promoter for soil and in the meantime, as an alternative nano pesticide, to combat pests infested maize grains thru post-harvest, in order to resolve the insect resistance to the conventional pesticides, which reflect the novelty of this work compared with the traditional relevance, for the pests control. The results concluded and assured that, by feeding the soil with silica nanoparticles up to 10 g/kg, the best growth and yield enhancement of maize crop is noticed. Moreover, the combination between mineral NPK and silica nanoparticles on soil application, had a beneficial effect on photosynthesis, yield enhancement and increased productivity of maize plants too. Also, silica nanoparticles displayed great success in combating the stored maize insects, which reached a 100% mortality rate.

In this current research work, we aimed to develop a new strategy for soil applications in terms of feeding or fertilizing, and at the same time, as a pesticide for combating the different kinds of pests that are found through storing maize grains. These results agree with Kyuma and Suriyaprabha et al. [49,50] and Sommer et al. [17] who showed the role of Si (Silicon) as a micronutrient for helping plants achieve the optimal use of water and other nutrients from soil. Also, was agreeing with [16] who detect the effect of NPK fertilizer on the yield and yield compounds in maize. 

The current results observe the same trend as [51], who presented that growth and yield characteristics were much influenced with increasing concentration of SiO_2_-NPs [51]. They observed that the physiological changes showed that the expression of organic compounds such as proteins, chlorophyll, and phenols, as well as the growth and yield of maize increased by using SiO_2_-NPs. Also, Farooq and Dietz (2015) showed the role of Silicon as a versatile player in plants [52]. 

The results are in accordance with Dung et al. (2016) that used SiO_2_-NPs in different doses and reported that 60 ppm caused an increase in fresh weight, dry weight, and chlorophyll content in chili plants [53]. Another study [54] reported that SiO_2_-NPs play a great role in the physiological components of maize, thus supporting the use of mineral fertilizers based on the distribution of roots and shoots. In addition, SiO_2_-NPs are essential in increasing the detailed functional properties of mineral fertilizers [54].

These results in an agreement with [55,56] whose reported that the application of Nano silica (8 g/L) showed significantly increased the growth traits of tomato plants [55,56]. Also, [57,58,59] presented that SiO_2−_NPs nutrition decreased the inhibitory outcome of salinity on plant growth by decreasing the Na+ content, nutrient uptake, increasing the cell wall peroxidase activities.

The results showing the efficiency of SiO_2_ in maize growth and productivity and these results were the same observation detected for SiO_2_-NPs that increased plant growth as reported by [55], and plant resistance to hydroponic conditions as reported by [51], as well as increased root length in plants, as stated by [60,61], and induced an improvement in photosynthesis as mentioned by [62].

Our results observed the same trend as other studies which showed the effects of Silica NPs with mineral fertilizers in many crop plants, such as maize as stated by [51,55], common bean as reported by [63], tall wheatgrass as described by [64], tomato as outlined by [65], faba bean as mentioned by [66], wheat as described by [67], rice as disclosed by [68], Glycine max as mentioned by [69], and sweet pepper as displayed by [70]. Also, others showed the effectives role of nanomaterial fertilizers on plant growth and productivity [21]. On the other hand, several research works have been carried out to prove the positive impact of silica nanoparticles to the crops, such as Rastogi et al. (2019) who reported the benefits of SiO_2−_NPs on physiological features of the plant in which that, they allow them to enter plants and affect its metabolic activities [71]. The same group also claim that the mesoporous nature of silica nanoparticles can also direct them to be good applicants as nano carriers for several molecules that may support in agriculture [71].

The current results showed that SiO_2_-NPs disclosed effectives against the stored products insects, which reached to 100% (M%). These data are in agreement with [41], [72,73], and [37] who reported the impact of nanomaterials as an alternative pesticide against stored grains insects. Our results designated that nanoparticles could help to produce new insecticides and this finding agreed with [74] who reported the same fact in addition to yield pesticides and insect repellants. Few researches have been carried out to study the toxicity effect of nanoparticles on insects especially storage insects, [75] stated that nanoparticles loaded with garlic essential oil is useful against *Tribolium castaneum* (Herbst) [75]. So, the use of nanoparticles as unconventional pesticide constitutes a new approach to combat pests which have become resistant to chemical conventional pesticides.

Silicon nanoparticles have enormous application as insecticides on different insects such as aphids,, cotton leaf worm, *Sitophilus oryzae* L., *Tribolium castaneum* (Herbst) and *Rhizopertha dominica* F under laboratory conditions [76]. The insect control mechanism is dependent on the structure of cuticular lipids for defending their water barrier, and in that way, prohibit death through dehydration. Meanwhile, silica nanoparticles get absorbed into the cuticular lipids by physio sorption and thus causes insects death. Moreover, Barik et al. [77] also verified the use of SiO_2−_NPs as a nano-pesticide and clarified the same control mechanism of combating insects

## 5. Conclusions

High throughput silica nanoparticles (SiO_2_-NPs) were synthesized via the sol–gel technique. The SiO_2_-NPs were obtained in a powder form followed by full characterization using state of the art analysis. TEM displayed that the particle shape was spherical with low disparity due to the cluster effect of silica particles. However, these aggregated particles are less than 50 nm. Dynamic light scattering (DLS) confirmed the particle size and stability, where the average size was around 68 nm. In addition, the zeta potential value of the prepared SiO_2_-NPs was −40 mV, which affirms the stability of these particles against aggregation, even after long time. Moreover, XRD ascertained that SiO_2_-NPs were prepared successfully using the sol–gel technique, in pure form and free from any other impurities or unreacted compounds.

To this end, the SiO_2_-NPs were applied successfully as both nanofertilizer and pesticide against four common pests that infect the stored maize and cause severe damage to them. The results obtained demonstrate that, by feeding the soil with SiO_2_-NPs up to 10 g/kg, the best growth and yield enhancement of maize crop is noticed. Mineral NPK interacted significantly with SiO_2_-NPs, whereas the application of mineral NPK at the rate of 50% with 10 g/kg SiO_2_-NPs, increased the highest mean values of agronomic features. Consequently, it can be concluded that the combination of mineral NPK and SiO_2_-NPs by soil application, had a beneficial effect on photosynthesis, yield enhancement, and increased the productivity of maize plants. In addition, it improved protein content (%) to 12.59% and chlorophyll content to (45.13 SPAD). This increase emphasizes the metabolic balance between induction of chlorophyll and proteins and cell wall transporters, damping off stress-responsive enzyme activities as a function of SiO_2_-NPs application in maize plants. Also, SiO_2_-NPs exhibited effectiveness against the stored products insects, which reached a 100% mortality rate. 

Finally, SiO_2_-NPs can be easily applied as growth promoter and can work as strong unconventional pesticides for crops during storage, with a very small and safe dose in order to combat all kinds of pests harmful to maize during storage.

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
