# Peer review of "Soil Application of Nano Silica on Maize Yield and Its Insecticidal Activity Against Some Stored Insects After the Post-Harvest"

_nanomaterials, 2020, doi:10.3390/nano10040739_

Round 1

Reviewer 1 Report

Manuscript ID: nanomaterials-755082, entitled ‘Soil Application of High-throughput Nano Silica on Maize Yield and its Insecticidal Activity Against Some Stored Insects After the Post-harvest’.

This manuscript should be of interest to the readers of the Nanomaterials Journal. However, the manuscript requires some work before it can be approved for publication. The language required improvement; a thorough review of the English is needed.

Specific comments

Title: the words high-throughput should be taken out from the title and in Conclusions, as the work has nothing to do with this. 

Introduction

Line 79: the sol-gel is ‘cheap’, this is compared to what and how cheap?

Line 81-89: many wrong words being used, bad sentences, all need to be changed.

Also if it so effective at killing pest, isn’t it harmful to the crops? some explanation on this and how they can be effective in controlling pests is needed.

Materials and methods

Line 119: define TEM when introduced for first time

Line 138 and everywhere: N/fed. Can you change this to SI unit?

Line 182: promoter??

Lines 190 & 216: scheme 1, words too small, not legible. Fig. 1, X and Y-axes labels too small.

Lines 296-319: many wrong vocabularies, need improvement.

Line 325-328: Figs. 7 & 8, again too small, not legible.

Discussion

Line 330: what is ‘very high concentration’?

Line 340 and beyond: discussion on how these nanoparticles can combat pest is needed. Cost-effectiveness of this technology also needs to be mentioned.

Author Response

"Please see the attachment." is the full revised file

Reviewer_1

Comments and Suggestions for Authors

Manuscript ID: nanomaterials-755082, entitled ‘Soil Application of High-throughput Nano Silica on Maize Yield and its Insecticidal Activity Against Some Stored Insects After the Post-harvest’. This manuscript should be of interest to the readers of the Nanomaterials Journal. However, the manuscript requires some work before it can be approved for publication. The language required improvement; a thorough review of the English is needed.

Specific comments

Title: the words high-throughput should be taken out from the title and in Conclusions, as the work has nothing to do with this. 

  • Response: Thanks for your comment. The manuscript title has been changed to: Soil Application of Nano Silica on Maize Yield and its Insecticidal Activity Against Some Stored Insects After the Post-harvest.

Introduction

Line 79: the sol-gel is ‘cheap’, this is compared to what and how cheap?

  • Response: In our work, we aimed to prepare high concentration of silica nanoparticles using the cheap precursor; TEOS in presence of ethanol. The whole experiment for the preparation of silica nanoparticles consumed cheap compounds. Thus, considering the price of the precursors, the preparation in its high concentration is not expensive via using such technique compared with other physical method used.

Line 81-89: many wrong words being used, bad sentences, all need to be changed.

  • Response: Thanks for your observation, the whole manuscript has been checked and the bad sentences have been corrected and rewritten.

Also if it so effective at killing pest, isn’t it harmful to the crops? some explanation on this and how they can be effective in controlling pests is needed.

  • Response: The prepared silica nanoparticles in its current form is not toxic since our aim in this current work was to prepare it in a pure form without any impurities, thus, calcination process has been carried out to degrade the undesired and unreactive compounds of TEOS or ammonia, ethanol substances. On the other hand, several research works have been carried out to prove the positive impact of silica nanoparticles to the crops, such as Rastogi et al. (2019) who reported the benefits of SiNPs on physiological features of the plant in which that, they allow them to enter plants and affect its metabolic activities. The same group claim also that, the mesoporous nature of silica nanoparticles can also direct them to be good applicants as Nano carriers for several molecules that may support in agriculture.
  • Moreover, Barik et al. (2008) verified consuming of SiNPs as nano-pesticide and clarified the control mechanism of combating insect which is more or less depending on the structure of cuticular lipids for defending their water barrier and in that way, prohibit death thru dehydration. 

Materials and methods

Line 119: define TEM when introduced for first time

  • Response: The TEM abbreviation has been identified, and amended by the word-track changes.

Line 138 and everywhere: N/fed. Can you change this to SI unit?

  • Response: All units are changed to SI unit according to the referee comment. Please find all changes in section: 2.1. and section 2.4. amended by the word-track changes.

Line 182: promoter??

  • Response: The word has been amended.

Lines 190 & 216: scheme 1, words too small, not legible. Fig. 1, X and Y-axes labels too small.

  • Response: scheme 1 is replaced by a new modified one with clear words. Also, Fig. 1, X and Y-axes labels are amended.

Lines 296-319: many wrong vocabularies, need improvement.

  • Response: The sentences have been rewritten, and amended by the word-track changes.

Line 325-328: Figs. 7 & 8, again too small, not legible.

  • Response: The figures have been replotted.

Discussion

Line 330: what is ‘very high concentration’?

  • Response: It means that compared with literature review, that we could prepare SiO2-NPs with high concentration.

Line 340 and beyond: discussion on how these nanoparticles can combat pest is needed. Cost-effectiveness of this technology also needs to be mentioned.

  • Response: The silicon nanoparticles has enormous applications as insecticides. Different insects such as aphids (El-Wahab et al (2016), cotton leaf worm (Ayoub et al. (2017); Sitophilus oryzae (L.), (Debnath et al. (2011) and El-Samahy et al. (2015) who used new silica nanoparticles formulation as stored product protectant in Sitophilus oryzae (L.), Tribolium castaneum (Herbst) and Rhizopertha dominica (F.) on wheat grains under laboratory conditions. The insect control mechanism is depending on a structure of cuticular lipids for defending their water barrier and in that way, prohibit death thru dehydration, while, silica nanoparticles gets absorbed into the cuticular lipids by physio sorption and thus causes insects death.
  • As mentioned before, the whole experiment for the preparation of silica nanoparticles used cheap materials. Thus, considering the price of precursors, the preparation in its high concentration is not expensive thru using such technique, hence, the use of this technology is considered as cost effective.

Reviewer 2 Report

The authors the application of nanosilica in maize crops and its insecticidal activity. The manuscript is well written and is of interest to the community. The article can be published after some minor grammatical corrections.

  1. The introduction needs to be more clearly written detailing some recent research efforts in this area.
  2. Control experiments need to be performed to claim the efficiency of nanosilica as insecticide, rather than just mention the different insects killed.

Author Response

" Please see the attachment is the full revised file"

Comments and Suggestions for Authors

The authors the application of nanosilica in maize crops and its insecticidal activity. The manuscript is well written and is of interest to the community. The article can be published after some minor grammatical corrections.

1-The introduction needs to be more clearly written detailing some recent research efforts in this area.  

  • Response: The introduction part has been updated with more relevant reference to be clearer for the readers.
  1. Control experiments need to be performed to claim the efficiency of nanosilica as insecticide, rather than just mention the different insects killed.
  • Response: the main objectives of this current research are dived into three key subjects: a) preparation and characterization of SiO2-NPs using the sol-gel method, b) evaluation of the influence of the combination between SiO2-NPs and mineral NPK, as soil application and their interaction with the plant characteristics of maize, c) application of SiO2-NPs as an alternative pesticides to combat pests infested maize grains thru post-harvest as well as to resolve the insect resistance to the conventional pesticides.

Reviewer 3 Report

The authors provide the results of using silica nanoparticles as part of fertilizers and as a pesticide. The study has a very robust design. The authors have a strong literature background in their field.

The paper needs corrections in language or signs. Examples:
line 50: plentiful arenas
line 78: expansively (where extensively would fit)
line 286: products (as a term fort he insects)
mv instead of mV in several places
subscripts in formulas; Cu-k-alpha
In part this might be due to automated formatting, but only in part. This must be checked by a technical editor of MDPI

Other points:

Lines 179-191 are not results, but another introductory part, please merge with introduction

Line 231  XRD is not a method to detect impurities, please reformulate

Tab.3  The effect of SiO2-NP is much weaker than that of NPK, compared to control. This is of coarse visible, but should be stated more clearly. The optimum mixture of blending NPK with SiO2-NP might be a statistical outlier, given the statistical error. This should be also clearly stated. Additional research and repeted tests are necessary in the future.

Fig. 1   does not transport scientific information, it would suit for public information. Is this intended?

Author Response

Please see the attachment is the full revised file 

Comments and Suggestions for Authors

  1. The authors provide the results of using silica nanoparticles as part of fertilizers and as a pesticide. The study has a very robust design. The authors have a strong literature background in their field.  
  • Response: Thanks for your positive comment.
  1. The paper needs corrections in language or signs. Examples:
    line 50: plentiful arenas
  • Response: The sentence has been amended.

line 78: expansively (where extensively would fit)

  • Response: Thanks for your comment, the word has been replaced.

line 286: products (as a term fort he insects) The word has been altered mv instead of mV in several places.

  • Response: Done as required by the reviewer.

subscripts in formulas; Cu-k-alpha

  • Response: Corrected
  1. In part this might be due to automated formatting, but only in part. This must be checked by a technical editor of MDPI
  • Response: I agree with your comment.
  1. Other points:

Lines 179-191 are not results, but another introductory part, please merge with introduction.

  • Response:Thanks for your comment, however, this paragraph has been added at the beginning of result and discussion part to highlight and outline the schematic image.

 Line 231, XRD is not a method to detect impurities, please reformulate

  • Response: The sentence has been changed to “Based on the aforementioned peaks, SiO2-NPs can be prepared successfully using sol-gel technique”.
  1. Tab 3  The effect of SiO2-NP is much weaker than that of NPK, compared to control. This is of course visible, but should be stated more clearly. The optimum mixture of blending NPK with SiO2-NP might be a statistical outlier, given the statistical error. This should be also clearly stated. Additional research and repeated tests are necessary in the future.
  • Respond: Thank you very much for your valuable comment., we will consider your advices in the next future work.
  1. Fig 1 does not transport scientific information; it would suit for public information. Is this intended?
  • Respond: The Scheme 1 you mentioned is just steps for the preparation and utilization of SiO2-NPs as feeding or fertilizing and in the same time, as pesticide for combating the different kinds of pests that are found thru storing maize grains. The resolution of this scheme is improved and inserted again in the entire text.
